# Evidence that birds sleep in mid-flight

Niels C. Rattenborg[1,*], Bryson Voirin[1,2,*], Sebastian M. Cruz[3], Ryan Tisdale[1], Giacomo Dell'Omo[4], Hans-Peter Lipp[5,6,7], Martin Wikelski[3,8] & Alexei L. Vyssotski[9]

Many birds fly non-stop for days or longer, but do they sleep in flight and if so, how? It is commonly assumed that flying birds maintain environmental awareness and aerodynamic control by sleeping with only one eye closed and one cerebral hemisphere at a time. However, sleep has never been demonstrated in flying birds. Here, using electroencephalogram recordings of great frigatebirds (*Fregata minor*) flying over the ocean for up to 10 days, we show that they can sleep with either one hemisphere at a time or both hemispheres simultaneously. Also unexpectedly, frigatebirds sleep for only $0.69\,\mathrm{h\,d^{-1}}$ (7.4% of the time spent sleeping on land), indicating that ecological demands for attention usually exceed the attention afforded by sleeping unihemispherically. In addition to establishing that birds can sleep in flight, our results challenge the view that they sustain prolonged flights by obtaining normal amounts of sleep on the wing.

[1] Avian Sleep Group, Max Planck Institute for Ornithology, Seewiesen 82319, Germany. [2] California Academy of Sciences, San Francisco, California 94118, USA. [3] Department of Migration and Immuno-Ecology, Max Planck Institute for Ornithology, Radolfzell 78315, Germany. [4] *Ornis italica*, Rome 00199, Italy. [5] Institute of Anatomy, University of Zurich, Zurich 8057, Switzerland. [6] Institute of Evolutionary Medicine, University of Zurich, Zurich 8057, Switzerland. [7] School of Laboratory Medicine and Medical Sciences, Department of Physiology, Kwazulu-Natal University, Durban 4000, South Africa. [8] Department of Biology, University of Konstanz, Konstanz 78457, Germany. [9] Institute of Neuroinformatics, University of Zurich and Swiss Federal Institute of Technology (ETH), Zurich 8057, Switzerland. * These authors contributed equally to this work. Correspondence and requests for materials should be addressed to N.C.R. (email: rattenborg@orn.mpg.de) or to A.L.V. (email: alexei@vyssotski.ch).

A diverse array of birds, including swifts[1–4], sandpipers[5,6], songbirds[7–10] and seabirds[11–13], engage in non-stop flights lasting several days, weeks, or longer. Given the adverse effects of sleep loss experienced by most animals[14] it is commonly assumed that birds fulfil their daily need for sleep on the wing[15]. However, the recent discovery that some birds can perform adaptively for several weeks despite greatly reducing the time spent sleeping[16] raised the possibility that birds forgo sleep altogether during long flights. Consequently, evidence of prolonged flights is not by default evidence of sleep in flight—neurophysiological recordings of the changes in brain activity that characterize sleep are required to answer this question. Furthermore, such recordings are needed to establish the amount, intensity, and types of sleep, and the potential implications that flight-related sleep adaptations have for understanding the functions of sleep. Due to the absence of recordings of brain activity during long flights, it is unknown whether birds sleep on the wing[15].

On land, birds can switch from sleeping with both hemispheres simultaneously to sleeping with one hemisphere at a time in response to changing ecological demands[17,18]. During such unihemispheric slow wave sleep (USWS) birds keep the eye connected to the awake hemisphere open and directed toward potential threats. Dolphins also use USWS to monitor their environment and can swim during this state[19]. Consequently, flying birds might rely on USWS to maintain environmental awareness and aerodynamic control of the wings, while obtaining the sleep needed to sustain attention during wakefulness. We tested this hypothesis in great frigatebirds (Fregata minor).

As Darwin observed during his voyage to the Galápagos Islands[20], frigatebirds are not known to rest on the water despite spending weeks to months flying over the ocean[12,13,21]. Their long wings, poorly webbed feet and reduced feather waterproofing make taking off difficult following more than momentary contact with the water. To catch food, great frigatebirds rely on large predatory fish and cetaceans to drive prey, such as flying-fish and -squid, to and above the surface[12]. Although previous studies detected potential feeding episodes (that is, slow flight near the surface) primarily during the day[12,21], under favourable conditions feeding also may occur at night[22], as frigatebirds follow ocean eddies predictive of foraging opportunities during the day and night[23]. Consequently, frigatebirds face ecological demands for wakefulness 24/7 while over the ocean.

By recording the brain activity of frigatebirds flying over the ocean, we demonstrate that they can sleep in flight with one hemisphere at a time or both together. Although frigatebirds engage in both types of sleep on the wing, sleep is more asymmetric in flight than when on land. Frigatebirds sleep mostly while circling in rising air currents and keep the eye connected to the awake hemisphere facing the direction of flight, suggesting that they use unihemispheric sleep to watch where they are going. Despite being able to sleep on the wing, when compared with sleep on land flying frigatebirds sleep very little, in shorter bouts, and less deeply, suggesting that frigatebirds face ecological demands for attention that usually cannot be met through sleeping unihemispherically. The ability to sustain cognitive performance on limited sleep challenges the dominant view that large daily amounts of sleep are required to maintain adaptive performance.

## Results

### Flight behaviour of frigatebirds
We used a data logging device (Neurologger 2A) designed for recording the electroencephalogram (EEG) of flying homing pigeons[24] combined with a three-dimensional (3D) accelerometer[25] to record brain activity and head movements in breeding female great frigatebirds (Fig. 1a) flying over the Pacific Ocean ($N = 14$) and after returning to their nest on Genovesa Island (Galápagos, $N = 9$). For each hemisphere, the EEG was recorded from the hyperpallium, a primary visual area (Fig. 1b). In addition, the birds' movements and altitude were recorded with GPS data loggers. All birds engaged in one or two trips over the ocean ($1.21 \pm 0.12$, s.e.m.) lasting up to 10 days ($5.76 \pm 0.67$ d, s.e.m.; range, 0.26–10.02 d; see Supplementary Figs 1–8) and spanning up to 3,000 km ($1988.45 \pm 186.33$ km, s.e.m.; range, 128.75–3001.42 km). Most birds completed a roughly clockwise loop over the ocean north-east of the Galápagos Islands (Fig. 1c). The birds spent less time flapping at night ($7.31 \pm 0.59\%$, s.e.m.) than during the day ($13.49 \pm 0.53\%$; $P = 6.0 \times 10^{-7}$; paired two-tailed Student's $t$-test; Supplementary Fig. 9a,c). The frigatebirds' altitude peaked in the hour before sunset and decreased across the night (Supplementary Fig. 9b). On average, the birds' altitude did not differ between the day ($137.9 \pm 4.7$ m, s.e.m.) and night ($136.5 \pm 3.8$ m; $P < 0.78$). Periods of potential foraging (flight below 20 m) occurred primarily during the day (Supplementary Fig. 9c). During the day and night the birds occasionally ascended ($1.45 \pm 0.15$ ascents per day, s.e.m.) to markedly higher altitudes ($905.9 \pm 25.4$ m for ascents $> 600$ m; maximum range 1013.6–1459.7 m; Supplementary Figs 9b and 10; see also Supplementary Discussion). As previously reported[13,21,26], the typical flight pattern of frigatebirds consists of circular rising on thermals (soaring) followed by straight gliding down (Fig. 1d, Supplementary Movie 1). These flight modes are reflected in the accelerometry recordings by centripetal acceleration (Fig. 1e; Methods). Circling and straight flight, as determined from a head-mounted accelerometer in great frigatebirds, were associated with slower and faster airspeeds, respectively ($5.40 \pm 0.23$ versus $8.59 \pm 0.21$ m s$^{-1}$, s.e.m., $P = 2.6 \times 10^{-8}$, paired two-tailed Student's $t$-test). Time spent circling increased across the day and decreased across the night ($P = 2.4 \times 10^{-6}$ and $P = 1.8 \times 10^{-7}$ for the respective linear trends, Supplementary Fig. 9c), likely reflecting diel variation in the availability of thermals. The birds circled to the left and to the right in equal amounts (Supplementary Fig. 11) and with equal centripetal acceleration ($P = 0.27$, paired two-tailed Student's $t$-test).

### Sleep in flight
The EEG patterns in flight were similar to those observed on land and in other birds. When gliding during the day, the EEG showed low-amplitude and high-frequency activity typical of alert wakefulness. In addition, frequent high-amplitude signals occurred in conjunction with rapid head movements, likely reflecting visual processing in the hyperpallium[27] during active searching for foraging opportunities. The rapid head movements and associated potentials in the EEG gradually disappeared and reappeared within the first hour after sunset and the hour before sunrise, respectively. During flapping flight at night, the EEG showed a pattern indicative of wakefulness in both hemispheres occasionally punctuated by isolated high-amplitude, slow waves (Fig. 2a). During flight without flapping, the wakefulness pattern usually persisted, but was occasionally replaced by continuous high-amplitude, slow waves (Fig. 2a). These slow waves were not correlated with fine head movements detected by the accelerometer, and were absent during much larger movements associated with flapping (Fig. 2a). Consequently, this EEG activity reflects slow wave sleep (SWS) rather than movement artifacts. During circling flight, the angle of the bill relative to the horizon increased slightly in SWS when compared with wakefulness ($P < 0.004$, paired two-tailed Student's $t$-test), perhaps as a result of the birds drawing their heads up and into the body during sleep, as observed on land

(Supplementary Fig. 12). On rare occasions, bouts of SWS were interrupted by brief episodes of apparent rapid eye movement (REM) sleep characterized by EEG activity in both hemispheres similar to alert wakefulness, dropping of the head, and twitching like that frequently observed during REM sleep on land (Fig. 2a, Supplementary Fig. 13; see also Supplementary Discussion). The birds typically ascended during SWS (rate of climb, $0.154 \pm 0.022\,\mathrm{m\,s^{-1}}$, s.e.m.) and descended during wakefulness (rate of climb, $-0.0046 \pm 0.0011\,\mathrm{m\,s^{-1}}$; $P = 9.1 \times 10^{-6}$, paired two-tailed Student's $t$-test), with SWS occurring at higher altitudes than wakefulness ($159.4 \pm 5.8\,\mathrm{m}$, s.e.m., and $135.3 \pm 4.0\,\mathrm{m}$, respectively; $P = 0.0017$, paired two-tailed Student's $t$-test; Supplementary Movies 2 and 3).

**Asymmetric sleep linked to circling flight.** The interhemispheric asymmetry in EEG slow wave activity (SWA; 0.75–4.5 Hz power) varied during SWS in flight. To quantify the frigatebirds'

utilization of USWS, we used an asymmetry index [AI = (left hemisphere SWA–right hemisphere SWA)/(left hemisphere SWA + right hemisphere SWA)] to categorized SWS as bihemispheric (BSWS; $-0.3 < \mathrm{AI} < 0.3$) or asymmetric (ASWS; $-0.3 \geq \mathrm{AI} \geq 0.3$), with an absolute $\mathrm{AI} \geq 0.6$ indicating USWS[28]. All types of SWS occurred in flight (Fig. 2a,b). The percentage of SWS consisting of ASWS was higher in flight ($71.57 \pm 3.96\%$, s.e.m.) than on land ($47.64 \pm 2.38\%$; $P < 0.002$, paired two-tailed Student's $t$-test), as was the percentage of ASWS consisting of USWS (flight, $47.27 \pm 5.30\%$; land, $24.96 \pm 2.26\%$; $P < 0.003$, paired two-tailed Student's $t$-test). Even though SWS was more asymmetric in flight, the presence of BSWS on the wing indicates that ASWS is not required to maintain the aerodynamic control of soaring or gliding flight and therefore likely serves other functions.

Due to the nearly complete crossing of input from the eyes, the eye opposite the more awake hemisphere is usually open during

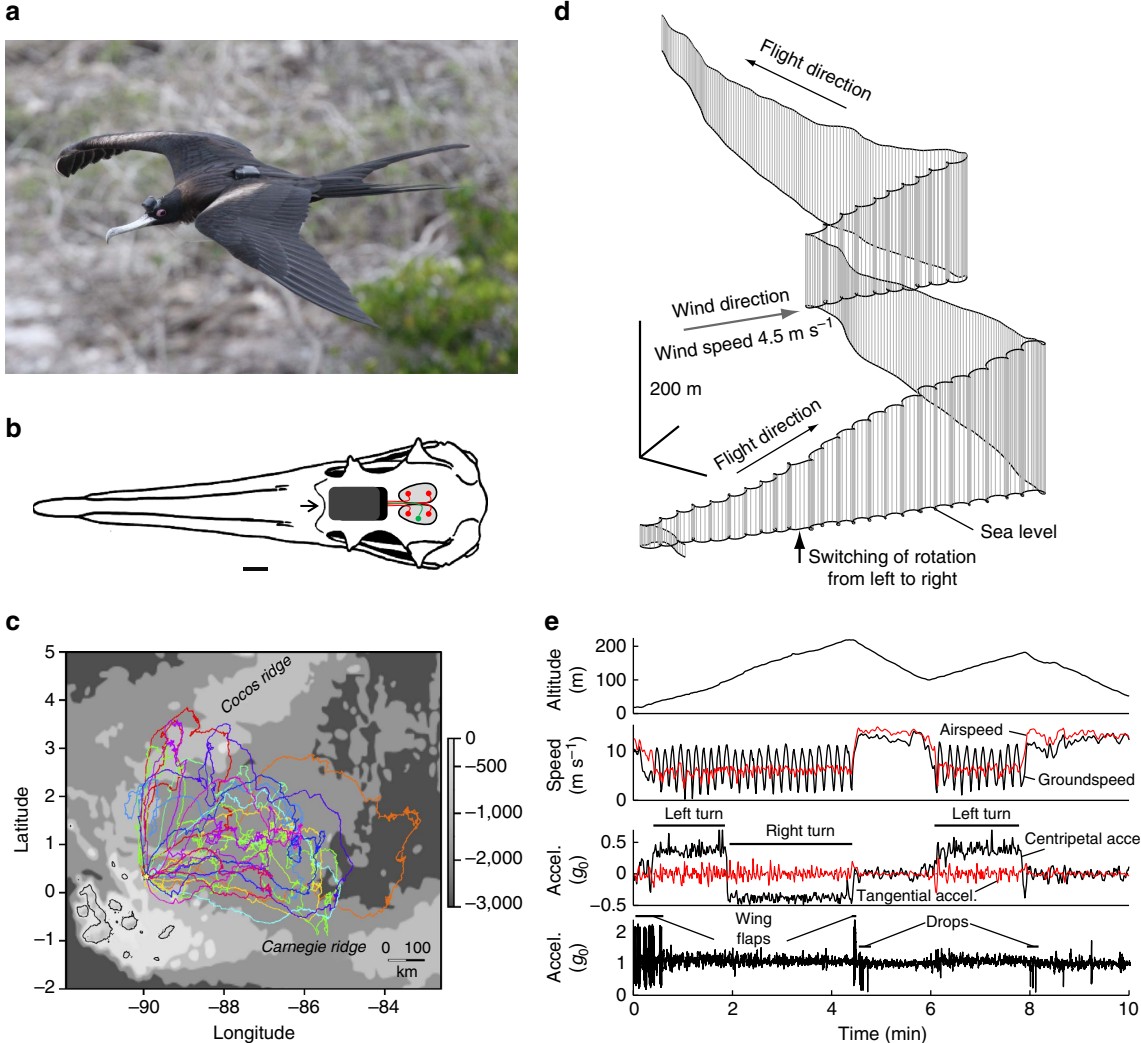

**Figure 1 | Measuring the brain state and flight mode of flying frigatebirds.** (**a**) Great frigatebird with a head-mounted data logger for recording the electroencephalogram (EEG) from both cerebral hemispheres and head acceleration in three dimensions. A back-mounted GPS logger recorded position and altitude. Photo: B.V. (**b**) Overhead view of a great frigatebird skull showing (1) the position of the cranial bulge (shaded grey) overlying the hyperpallium of each hemisphere, (2) the position of the epidural electrodes (red dots, EEG; green dot, ground) and (3) the data logger (black rectangle) just posterior to the naso-frontal hinge (arrow). Scale bar is 10 mm. (**c**) All GPS tracks for individual birds coded with different colours. The Galapagos Islands are outlined with black lines and the study site (Genovesa) is marked by a star. Ocean depth (m) is coded with grey scale. (**d**) High temporal resolution (1 Hz) 10 min flight trajectory recorded with GPS from a frigatebird (see Supplementary Movie 1 for 3D visualization) showing the circling (soaring) and straight (gliding) flight modes typical of Fregatidae[13] (Methods). (**e**) Altitude, ground speed and airspeed (computed from the GPS data in (**d**)), tangential and centripetal (radial) low-pass filtered acceleration, and the absolute value of total acceleration (measured by an accelerometer) for the flight in (**d**).

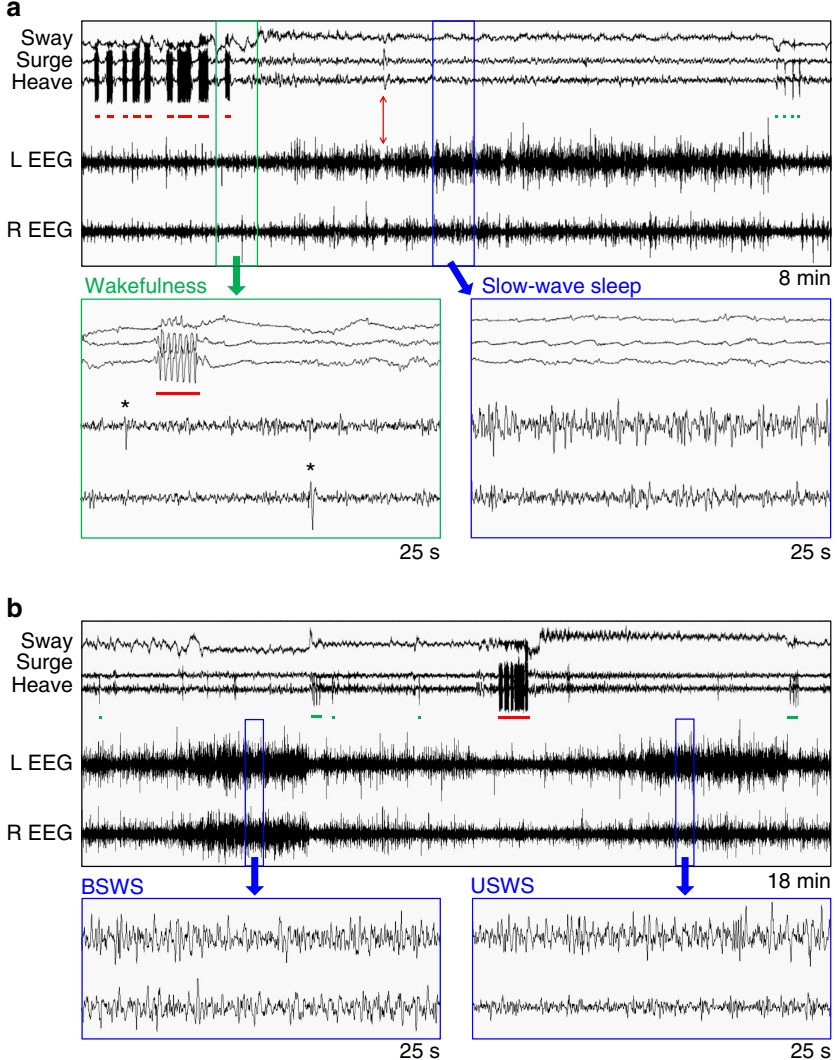

**Figure 2 | Unihemispheric and bihemispheric sleep in flight.** (**a**) Recording of head acceleration in three dimensions (sway, surge and heave) and electroencephalogram (EEG) activity from the left (L) and right (R) hemispheres showing the transition from wakefulness to SWS following the cessation of flapping (red bars). Brief episodes of dropping (green bars) occur after this episode of sleep. Expanded views (bottom) show wakefulness characterized by low-amplitude, high-frequency EEG activity in both hemispheres, infrequently punctuated by isolated high-amplitude, slow waves (*), and SWS characterized by continuous high-amplitude, slow waves, in this case, primarily in the left hemisphere. The red arrow (top) marks an episode of apparent REM sleep (expanded in Supplementary Fig. 13). (**b**) Recording from the same bird showing an episode of bihemispheric SWS (BSWS) and unihemispheric SWS (USWS), including expanded views of both states. The mean duration of episodes of sleep was shorter than these long episodes used to demonstrate USWS and BSWS in flight. These recordings are from frigatebird 13 (Supplementary Fig. 7).

ASWS[18], allowing birds to simultaneously sleep and watch for threats[17]. In frigatebirds, the accelerometer recordings suggest that ASWS serves a similar function in flight. During wakefulness and SWS on land, frigatebirds kept their head straight most of the time ($85.91 \pm 3.19\%$, s.e.m., and $89.67 \pm 2.18\%$, respectively, $P = 0.056$, paired two-tailed Student's $t$-test; Fig. 3a), as indicated by the distribution of sway axis values with a cluster around zero; head position was classified as straight when acceleration along the sway axis fell between the dashed vertical lines in Fig. 3a ($-0.175g_0$ and $0.175g_0$; standard acceleration of free fall $g_0 = 9.80665 \, \mathrm{m\,s^{-2}}$). In flight, acceleration along the sway axis also showed a unimodal peak around zero during wakefulness ($75.42 \pm 2.05\%$, s.e.m.). However, in contrast to SWS on land, the distribution of sway axis values was tri-modal during SWS in flight, with one peak around zero and two additional peaks reflecting acceleration to the left and right (Fig. 3a). Acceleration to the left and right was primarily due to radial

acceleration of the birds as they turned in either direction (wing angle, $18.75 \pm 0.48°$, s.e.m.; $P = 6.4 \times 10^{-15}$, paired two-tailed Student's $t$-test; Fig. 3b), likely reflecting soaring on rising air currents[13,21,26]. Interestingly, during ASWS the birds were more likely to circle toward the side with greater SWA (ASWS-L, to left, $65.31 \pm 5.07\%$, s.e.m., to right, $8.76 \pm 1.46\%$, $P = 1.1 \times 10^{-7}$; ASWS-R, to left, $9.31 \pm 1.81\%$, to right, $66.59 \pm 6.47\%$, $P = 1.2 \times 10^{-6}$, paired two-tailed Student's $t$-test), whereas during BSWS there was no bias for circling toward one particular side (to left, $35.22 \pm 4.31\%$, to right, $35.35 \pm 4.83\%$, $P = 0.73$, paired two-tailed Student's $t$-test; Fig. 3c,d). In addition to asymmetries in SWA, during SWS we also detected smaller asymmetries in gamma activity (30–80 Hz power), a frequency implicated in visual attention[29]. Opposite to SWA, during SWS with asymmetric gamma ($-0.1 > \mathrm{AI} > 0.1$) the birds accelerated toward the side with lower gamma (left gamma > right gamma, to left, $11.71 \pm 2.02\%$, s.e.m., to right, $65.76 \pm 6.21\%$, $P = 6.5 \times 10^{-6}$,

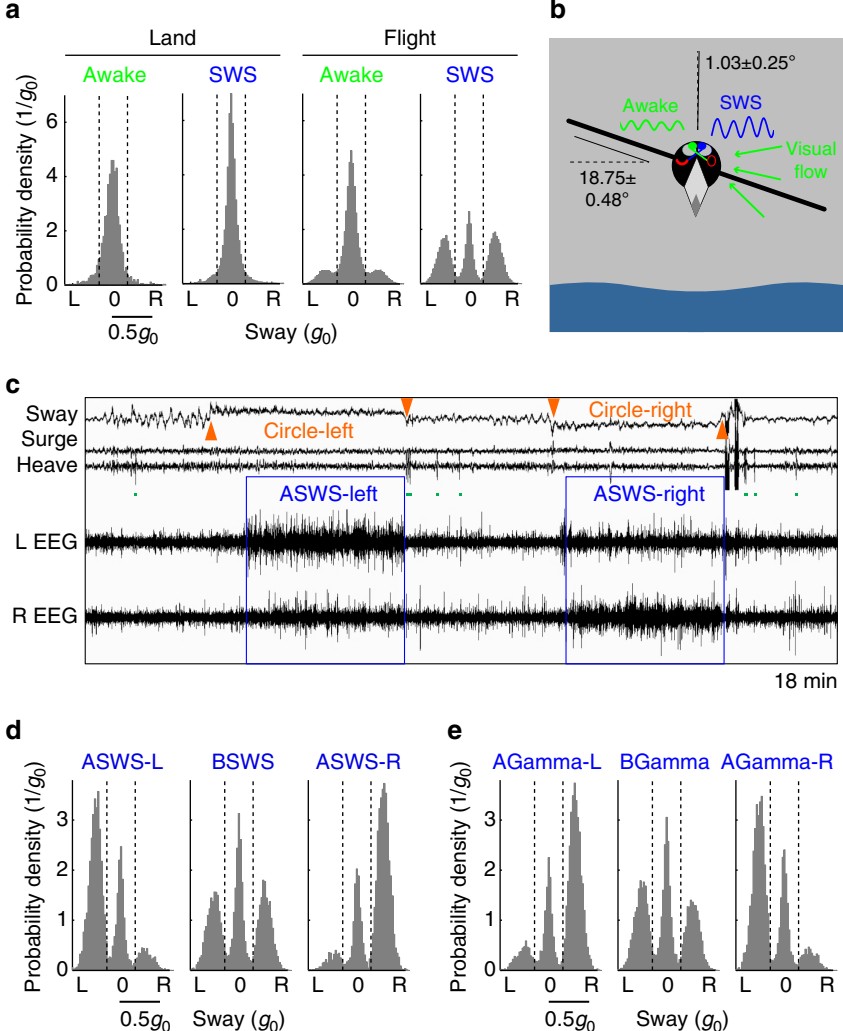

**Figure 3 | Slow wave sleep electroencephalogram (EEG) asymmetry is related to circling flight.** (**a**) Distribution of awake and SWS 4 s epochs (all birds combined) occurring at different sway accelerations ($0.02g_0$ bins) on land and in flight at night. On land (top), the values were clustered around zero while awake and in SWS indicating that the birds held their head straight during both states. Although the birds also held their head and wings straight while awake in flight, SWS, in most cases (70.57%), occurred with circling flight to the left and right, as reflected by sway acceleration $< -0.175g_0$ and $> 0.175g_0$ (dashed vertical lines). (**b**) Diagram showing the wing and head angle relative to the horizon during circling flight to the left calculated from the accelerometry. The corresponding brain state (see below) is also shown. (**c**) Recording showing the relationship between asymmetric SWS (ASWS) and acceleration along the sway axis; during ASWS with greater EEG slow wave activity (SWA; 0.75–4.5 Hz power) in the left hemisphere (ASWS-L) the sway axis showed high values corresponding to circling to the left, and when SWA was greater in the right hemisphere (ASWS-R), the sway axis showed low values corresponding to circling to the right. Same bird as in Fig. 2. (**d**) The relationship between sway acceleration and type of SWS in flight for all birds combined. Data from (**a**) are partitioned according to the type of SWS as defined in the main text; ASWS-L, ASWS-R and bihemispheric SWS (BSWS). (**e**) Relationship between flight mode (sway acceleration) and SWS in flight for data from (**a**) partitioned according to the interhemispheric asymmetry in gamma activity (30–80 Hz power); asymmetric gamma with greater gamma in the left (AGamma-L; $AI \geq 0.1$) or right (AGamma-R; $AI \leq -0.1$) hemisphere and bihemispheric (symmetric) gamma (BGamma; $-0.1 < AI < 0.1$). The overall relationship between circling flight, brain state and probable eye state[18] is summarized in (**b**); awake hyperpallium (green) and sleeping hyperpallium (blue) and the corresponding relative difference in EEG SWA. The green arrows show the general direction of visual flow while circling to the left.

paired two-tailed Student's *t*-test; left gamma < right gamma, to left, $62.47 \pm 6.46\%$, to right, $8.58 \pm 1.44\%$, $P = 1.2 \times 10^{-6}$, paired two-tailed Student's *t*-test) whereas during SWS with symmetric gamma there was no bias for acceleration toward one particular side (to left, $37.50 \pm 3.74\%$, to right, $31.77 \pm 4.20\%$, $P = 0.96$, paired two-tailed Student's *t*-test; Fig. 3e). The more awake EEG activity (that is, lower SWA and higher gamma) in the hemisphere opposite the direction of the turn indicates that the frigatebirds had the eye toward the direction of the turn open (Fig. 3b), presumably to watch where they were going.

**Sleep loss in flight**. The amount, timing, continuity and depth of sleep on the wing also suggest that frigatebirds face ecological demands for wakefulness throughout the day and night. In flight, frigatebirds slept for only $2.89 \pm 0.58\%$ (s.e.m.) of the time, whereas on land $53.28 \pm 4.82\%$ of the time was spent sleeping ($P = 1.1 \times 10^{-5}$, paired two-tailed Student's *t*-test; Fig. 4a; see Supplementary Fig. 14a for $N = 14$ in flight). Sleep occurred almost exclusively at night in flight (day, $0.36 \pm 0.16\%$, s.e.m.; night, $5.44 \pm 1.03\%$; $P = 2.4 \times 10^{-5}$, paired two-tailed Student's *t*-test), but throughout the day and night on land (day, $47.90 \pm 4.95\%$; night, $53.76 \pm 6.72\%$; $P = 0.45$, paired two-tailed

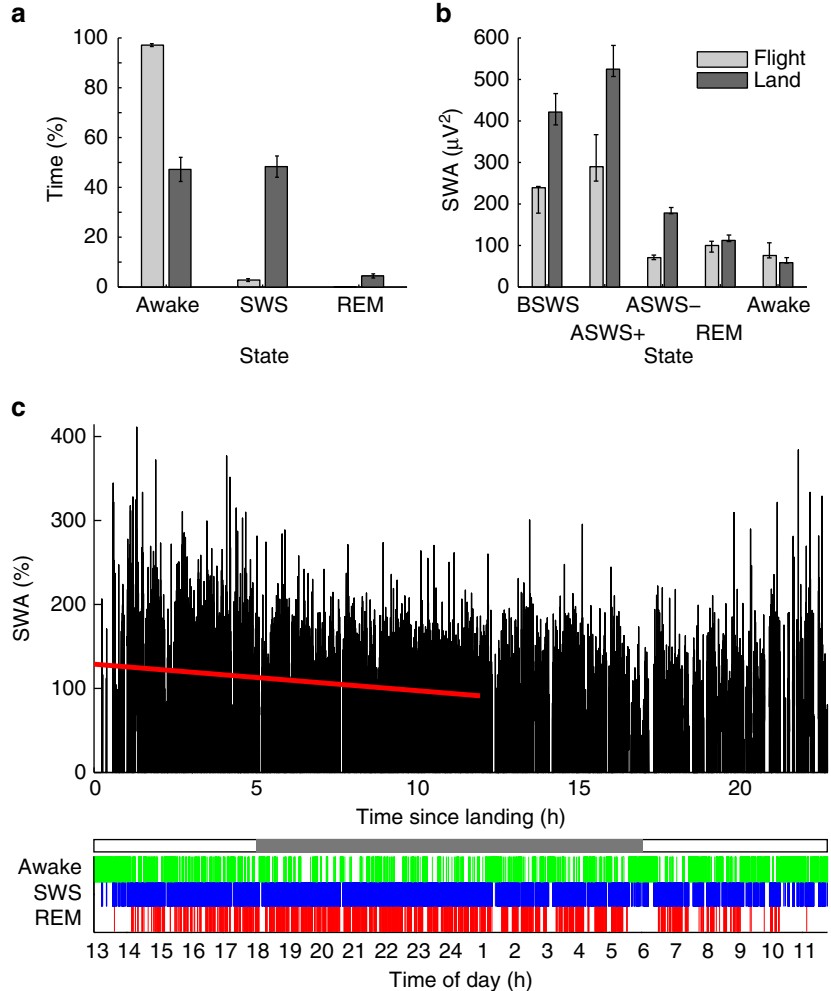

**Figure 4 | Frigatebirds sleep more and deeper on land than in flight.** (**a**) Time spent awake and in SWS and REM sleep in flight and on land. (**b**) Electroencephalogram (EEG) slow wave activity (SWA; 0.75–4.5 Hz power) while awake and in SWS and REM sleep in flight and on land (median and quartiles for the median). For SWS, SWA is shown for, (1) bihemispheric SWS (BSWS), (2) asymmetric SWS (ASWS) for the hemisphere with greater SWA (ASWS + ) and (3) ASWS for the hemisphere with lower SWA (ASWS–). (**c**) Decline in SWS-related SWA during the first 12 h since landing (top) and the corresponding sleep staging (bottom) in one bird; awake (green), SWS (blue), and REM sleep (red). In the photoperiod bar (middle) grey reflects night.

Student's *t*-test; Supplementary Fig. 9d–f). Even while gliding and soaring at night (that is, no wing flaps or drops), when sleep was possible, wakefulness encompassed 79.71 ± 1.34% (s.e.m.) of the 9.66 ± 0.16 h (s.e.m.; $N = 14$) spent in these flight modes. Episodes of SWS were also longer on land (28.25 ± 1.75 s, s.e.m.) than in flight (10.89 ± 0.81 s; $P = 2.24 \times 10^{-5}$, paired two-tailed Student's *t*-test), as was the maximum duration of SWS episodes (land, 272.89 ± 44.47 s, s.e.m., and flight, 134.22 ± 21.72 s, $P = 0.023$, paired two-tailed Student's *t*-test; maximum range: land, 128–572 s and flight, 48–216 s). With REM sleep included, episodes of sleep (SWS + REM sleep) were even longer on land (52.22 ± 4.91 s, maximum range, 204–1,212 s) than in flight (11.64 ± 1.00 s, $P = 4.1 \times 10^{-5}$, paired two-tailed Student's *t*-test; maximum range, 52–216 s); the longest episode of continuous sleep in flight (348 s) occurred in one of the five birds only recorded in flight. Episodes of SWS were longer during circling flight (12.55 ± 0.80 s, s.e.m.) than straight flight (6.69 ± 0.28 s, $P = 1.2 \times 10^{-4}$, paired two-tailed Student's *t*-test). The mean duration of REM sleep episodes in flight (4.92 ± 0.15 s, s.e.m.) was shorter than on land (5.92 ± 0.33 s; $P = 0.014$, paired two-tailed Student's *t*-test), and REM sleep as a percentage of total sleep time was lower in flight (3.52 ± 0.92%) than on land (8.15 ± 1.44%; $P = 0.0084$, paired two-tailed Student's *t*-test).

Finally, the intensity of SWS, based on EEG SWA, was lower in flight than on land during BSWS ($P = 0.02$, paired two-tailed Student's *t*-test) and for both hemispheres during ASWS (hemisphere with greater SWA, $P < 0.02$; hemisphere with lower SWA, $P = 0.008$, paired two-tailed Student's *t*-test; Fig. 4b; see Supplementary Fig. 14b for $N = 14$ in flight). Even on the last day of flight, when sleep pressure should have been the greatest, SWA was lower than on land and unchanged from earlier in the flight (Supplementary Fig. 15a,b). Collectively, the sleep patterns observed in flight indicate that in addition to the attention required for foraging during the day, frigatebirds face ecological demands for attention at night, as well as limits on the depth of sleep.

The higher amount of wakefulness, lower intensity of sleep and absence of an increase in sleep intensity across the flight question whether the homeostatic process that normally regulates sleep duration and intensity[30] in other birds is absent in frigatebirds or just suppressed during flight. On land, SWS-related SWA (both hemispheres combined) declined as a function of time since landing ( $-3.91 \pm 0.74\%$ per hour, s.e.m.; $P = 0.003$, two-tailed Student's *t*-test; Fig. 4c), as did the time spent in BSWS and ASWS ($P = 0.01$ and $P = 0.01$). Because the time course of both states did not differ ($P = 0.54$), the decline in total SWA reflects a

change in sleep intensity, rather than a change in the ratio of ASWS to BSWS across this period. The decline in sleep intensity after returning to land suggests that the homeostatic mechanisms present in other birds are also present in frigatebirds. Interestingly, this also indicates that frigatebirds are able to forestall these mechanisms when sleeping in flight. Finally, although frigatebirds recover lost sleep once back on land, recordings spanning the cycle between flights and time on land are needed to fully characterize the homeostatic process and determine whether frigatebirds on land compensate entirely for sleep lost in flight.

## Discussion

Several ecological factors may account for the characteristics of sleep in flight. Even though the frigatebirds rarely came near the surface at night[12,13,21], the low amount of sleep may be related to foraging. As during the day, great frigatebirds follow ocean eddies at night[23] to position themselves near potential foraging sites at daybreak and, perhaps, under favourable conditions, to forage at night[22]. Although it is unknown whether frigatebirds pay attention to atmospheric, olfactory, or visual cues to monitor the ocean at night[23,31], the low amount of sleep suggests that this task requires full attention exceeding that possible during USWS. Even during the little sleep that did occur in flight, frigatebirds sacrificed sleep for vigilance, as indicated by the greater degree of SWS asymmetry and the preference for keeping the eye connected to the awake hemisphere facing the direction of flight. Although the risk of falling in the water is reduced by their preference for sleeping in rising air currents and at higher altitudes, frigatebirds may still need to watch where they are going to avoid collisions with other birds. In this regard, even though flying frigatebirds have no predators, their utilization of ASWS is similar to that in ducks (*Anas platyrhynchos*) sleeping on land which direct the open eye toward a predatory threat[17].

Despite marked ecological and environmental differences, the ability to greatly reduce the time spent sleeping in flight is similar to that recently described in polygynous male pectoral sandpipers (*Calidris melanotos*)[16]. While breeding under the constant light of the Arctic summer, some males sleep very little during a 3-week period of male–male competition for mates. Interestingly, the males who sleep the least sire the most offspring, suggesting that resistance to the adverse effects of sleeplessness is under sexual selection. Our findings in frigatebirds demonstrate that other ecological pressures can also favour an ability to sustain wakefulness[32,33] even in animals living under equatorial photoperiods.

In contrast to frigatebirds, in humans[14] and other animals[34], including some birds[30,35], the adverse effects of sleep loss (for example, sleepiness and reduced attention) manifest rapidly and accumulate across days of sleep restriction. In addition to selection for resistance to the adverse effects of sleep loss, these divergent results might be explained by differences in motivation and associated brain neurochemistry[36]. The small amounts of sleep in flying frigatebirds may also serve as restorative 'power naps' that help them forestall the recovery of sleep until they return to land. Determining how flying frigatebirds sustain performance on little sleep may provide new perspectives on our understanding of the adverse effects of sleep loss experienced in humans.

## Methods

**Animals and instrumentation.** The Galápagos National Park Service approved of and granted the research permits for this work. During August, 2014, adult females ($N = 15$) caring for chicks on the coast of Darwin Bay, Genovesa Island, Galápagos, Ecuador (0°19′5.57″N, 89°57′1.23″W) were caught by hand on their nest at night. The chick was covered to keep it warm and safe while its mother was instrumented.

Using isoflurane anaesthesia and aseptic methods[16], for each cerebral hemisphere, EEG sensors were placed on the dura overlying the anterior (A) and posterior (P) hyperpallium, a structure that forms a pronounced bulge in the cranium of frigatebirds (Fig. 1b; for a CT scan of a similar skull see, www.digimorph.org/specimens/Fregata_magnificens/); the sensors were 8 mm apart along the AP axis, spanning the most pronounced portion of the cranial bulge, and 4 mm from the midline (Fig. 1b). A fifth sensor was placed laterally on the left hemisphere for the electrical ground. The gold-plated, round-tipped (0.5 mm diameter) sensors were secured with a small amount of dental acrylic cured with an ultraviolet light (Clearfil SE Bond, Kuraray Noritake Dental, Japan and Tetric EvoFlow, Ivoclar Vivadent AG, Schaan, Liechtenstein) and connected to a flexible, insulated spring wire (no. 276-0146-001; DSI, St. Paul, MN). The wires were soldered to a data logger (Neurologger 2A; www.evolocus.com, see also www.vyssotski.ch/neurologger2) which included a 3.6 V lithium battery (Saft LS-14250; www.saftbatteries.com) and a three-axis accelerometer (LIS302DLH; STMicroelectronics). The logger was glued (Hystoacryl, Aesculap AG, Germany and Pattex, Repair Gel, Henkel AG & Co. KGaA, Germany) to the skin and feathers just posterior to the naso-frontal hinge (Fig. 1b). The logger was configured to record bipolar EEGs from the left and right hemispheres, and acceleration in the three cardinal directions continuously at 200 Hz for up to 10 days. A GPS data logger (i-gotU, GT-600; www.i-gotu.com) configured to record position every 5 min was attached to the back feathers with gaffer tape (tesa, no. 4671; www.tesatape.com). The total weight (55 g) of the equipment was 4.0% of the birds' weight (1366.79 ± 24.09 g, s.e.m.). Fourteen of the 15 birds were recaptured 7.79 ± 0.49 d (s.e.m.; range, 5.37–10.45 d) later, after returning from at least one foraging trip. In nine of the birds, we obtained recordings (16.40 ± 3.33 h, s.e.m., in duration; Supplementary Fig. 9g) after they returned to the nest to evaluate sleep on land. At the end of the study, the equipment was removed under anaesthesia and the birds were released. On release, the birds resumed nesting behaviour indistinguishable from that observed in undisturbed birds. Finally, to validate our analysis of the flight trajectories in great frigatebirds, we used data recorded from two magnificent frigatebirds (*Fregata magnificens*) in a pilot study conducted in the French Guiana using a GPS data logger (GiPSy-2, www.technosmart.eu) with a 1 Hz sampling rate combined with a 3D acceleration logger (25 Hz rate; AXY-1, www.technosmart.eu; Fig. 1d,e).

**Sleep scoring and EEG analysis.** During flight, all days with stable EEGs were scored for time spent awake, and in SWS and REM sleep using 4 s epochs and REMLogic software (Natus Medical, Pleasanton, California)[16]. All recordings after returning to land were also scored, including the short landings between two flights observed in birds 1 and 5 (Supplementary Figs 1 and 3). A bout of a given state was defined as one or more epochs of that state uninterrupted by a single epoch of another state. The bout durations for wake, SWS, REM sleep and the overall amount of time spent in each state were based on all scored days. The spectral analysis of the EEG focused on a night with comparatively large amounts of sleep and high signal quality (see Supplementary Figs 1–8). For each state, all 4 s artifact free epochs were analysed with the fast Fourier transform (0.25 Hz bins) applied to Hamming-windowed data. SWA and gamma power were estimated from Fourier coefficients taken for ranges 0.75–4.5 and 30–80 Hz, respectively. Medians of SWA and gamma power were used for statistical comparisons. Quartiles for group comparisons shown in Fig. 4b and Supplementary Figs 14b and 15b are estimated by bootstrap. Interhemispheric asymmetries in SWA and gamma, and their relationship with the mode of flight (Fig. 3d,e), were based on the night with large amounts of sleep. In addition, SWS-related SWA was calculated for the last night of flight to detect potential changes in sleep intensity across the flight (Supplementary Fig. 15b).

**Accelerometry analysis.** The accelerometer recordings revealed two predominant patterns during flight (Fig. 2a). Flapping flight was characterized by large sinusoidal oscillations ($\approx 2.5$ Hz) in the heave and surge axes corresponding to individual wing beats. In contrast, during gliding and soaring flight, the three axes were largely flat or showed slow oscillations likely reflecting a combination of fine manoeuvres and respiratory movements (see expanded view for SWS in Fig. 2a). When gliding and soaring during the day, small, frequent and rapid horizontal movements of the head were superimposed on these slow oscillations. Flight was occasionally interrupted by a rapid decrease in acceleration along the heave axis, corresponding to the bird dropping, presumably due to momentary folding of the wings (Supplementary Movie 4). Finally, bouts of high-frequency activity occurred infrequently in all axes simultaneously, likely reflecting preening, as observed in birds flying over the colony and while on the nest.

Previous studies[12,13,26] and our own observations (Fig. 1d), show that frigatebirds exhibit two major flight trajectories; rising in circles (soaring) and straight gliding down. In addition to identifying flapping flight, the accelerometer was useful for discriminating circular from straight flight (Fig. 1e). During both types of flight the absolute air-referenced flight speed averaged over significant time intervals ($>4$ s) is constant (Fig. 1e). Thus, the tangential (co-directed with the speed vector) acceleration is zero in both flight modes. When the animal flies straight the total acceleration felt by the accelerometer is produced only by the gravity vector $g$ (standard gravity, $1g_0 = 9.80665$ m s$^{-2}$). However, during circular flight additional centripetal (radial) acceleration, $a_r = V^2/R$ ($V$, speed; $R$, radius of the trajectory) is added to the acceleration of gravity: $\vec{a}_{tot} = \vec{g} + \vec{a}_r$. As rotation lies

approximately in the horizontal plane, the two acceleration vectors are orthogonal to each other and total acceleration, $a_{tot} = \sqrt{g^2 + a_r^2}$. Thus, to determine whether the trajectory is straight or not, it is sufficient to measure total acceleration, low-pass filter it to remove the influence of wing flapping and compute radial acceleration from this equation. Radial acceleration above $0.175g_0$ corresponded to circling flight (Fig. 3, Supplementary Fig. 16). Total acceleration in circling flight was $1.057 \pm 0.003g_0$ and radial acceleration was $-0.340 \pm 0.009g_0$ (s.e.m.; see Supplementary Table 1 for values for individual birds). The bank (wing) angle during soaring was measured as $\arccos(g/a_{tot})$ and was $18.75 \pm 0.48°$ (s.e.m.). However, for our EEG analysis it was also important to know whether the bird was rotating to the right or to the left. This information was obtained by measuring radial acceleration with the accelerometer mounted on the bird's head with one axis (that is, sway) directed laterally. Because frigatebirds keep their heads straight during both flight modes, we were able to determine radial acceleration directly from the accelerometer without additional transformations. However, to confirm this claim and to increase the accuracy of the radial acceleration measurements we also performed computations without this assumption. The accelerometer was attached to the bird's head in a way such that one axis was orthogonal to the tangential plain of the bird skull and another was directed laterally. Projection of total acceleration on to the tangential plain of the bird skull clearly shows three clusters corresponding to straight and circling flight, with turning to the left and right (see data from one example bird in Supplementary Fig. 16a). To simplify this analysis, we rotated the axes of the head-fixed coordinate system to have one axis directed to the ground during straight flight; however, in the recording examples shown in Figs 2 and 3, Supplementary Fig. 13, acceleration is shown in the original axes of the accelerometer. The following analysis shows that the skull surface tangential plane deviated by $29.86 \pm 0.68°$ (s.e.m.) from the horizon (see Supplementary Table 1; see also Fig. 1a). As a first step we down-sampled the acceleration data to 25 Hz (from original 200 Hz) to decrease computation time. We then filtered out high frequencies by applying a low-pass finite impulse response filter (0.1 Hz; span 40 s). The input data were processed both in the forward and reverse directions and the resulting sequence had precisely zero-phase distortion and doubled filter order. Then, we computed principle components (PCs) in 3D space without mean subtraction. The first PC pointed in the direction of the gravity vector, the second—in the lateral (radial) direction, and the third—in the direction of the speed vector. Because we found that accuracy of the PCs determination can be affected by outliers, mainly due to episodes when the bird drops down with acceleration in the direction of the first PC $<0.95g_0$, we excluded such points and recomputed the PCs again. In the horizontal plain of the second and the third PCs (Supplementary Fig. 16b), clusters corresponding to rotation to the left and right were aligned relative to the coordinate axes. The best separation was observed along the second PC corresponding to sway acceleration. The vertical lines drawn at sway accelerations $\pm 0.175g_0$ reliably separate the three clusters in all birds. Because we wanted to compute rotations of the head relative to straight flight, we repeated the PC analysis, but for points representing straight flight only. Coordinates of the first PC gave the direction to the ground during straight flight. The angle between this direction and skull surface normal is the skull angle shown in Supplementary Table 1. We rotated the coordinate system a second time to have one axis in the direction of the first PC (Supplementary Fig. 17). In this head-fixed coordinate system, during circling flight, the absolute value of lateral (sway) acceleration was $0.321 \pm 0.008g_0$ (s.e.m.), acceleration in the direction of the flight (surge) was $0.028 \pm 0.005g_0$ and vertical (heave) acceleration was $1.006 \pm 0.001g_0$ (see Supplementary Table 1). Assuming zero tangential acceleration as before, we computed the angle of the head turn in circling flight ($2.137 \pm 0.184°$, s.e.m.) relative to straight flight and the direction of the axes over which the turn was performed (right–left: $0.626 \pm 0.096°$, beak–tail: $0.521 \pm 0.111°$, down–up: $-0.209 \pm 0.033°$, signs are valid for the case when the animal turns left, but absolute values represent averaged quantities for left and right turns taken together, see Supplementary Table 1). To simplify interpretation of the head turn we computed angular deviations of the head-fixed vector pointing upwards in the lateral (right–left) and anterior–posterior (beak–tail) directions. These deviations were $1.033 \pm 0.252°$ and $1.444 \pm 0.233°$ (signs correspond to the left turn as before). As shown in the table, bank angle (wings-to-horizon) was computed with the assumption that total acceleration was orthogonal to the plane of the wings. This assumption was verified by placing accelerometers on the backs of two magnificent frigatebirds together with the GPS logger in a pilot study (Supplementary Fig. 18). In these two birds, total acceleration during circling flight was 1.053 and $1.067g_0$. Standard deviations of sway acceleration were 0.013 and $0.016g_0$, and standard deviations of surge acceleration were 0.033 and $0.036g_0$, respectively. Thus, the standard deviation of the total acceleration vector in the lateral direction was $0.71°$ and $0.85°$ and in the anterior–posterior direction it was $1.80°$ and $1.38°$. Taking the 95% confident border as a more conservative estimate, we obtained $1.45°$ and $1.67°$ for sway and $3.60°$ and $2.81°$ for surge. These angles are much smaller than the angle of the wing plane to the horizon ($18.32°$ and $20.41°$). Thus, our assumption about orthogonality of the plane of the wings to total acceleration is correct.

**Detection of wing flaps and drops.** Wing flaps and drops were detected by analysing the absolute values of the acceleration vectors recorded by the accelerometer. As a first step, acceleration was down-sampled to 50 Hz to decrease computation time. Then the signal was band-pass filtered 0.25–5 Hz. The finite

impulse response filter with an 8 s span was applied in forward and reverse directions to ensure a zero time shift. Deviations in acceleration below $-0.4g_0$ were selected as potential flaps and drops. Flaps and drops were separated from noise and sorted by the shape of acceleration signal around these events ($\pm 0.64$ s). The 64-point fragments of the record centred around the detected acceleration minima were sorted using wavelets and a superparamagnetic clustering algorithm[37] (WaveClus 2.0 package for Matlab) in birds 1 and 2. After validating the classification algorithm and cluster matching in two birds, the recorded fragments from the remaining birds were sorted using a faster and simpler nearest neighbour algorithm (computing and comparing distances from non-classified elements to the members of the clusters already classified in bird 1). The average shapes of acceleration around flaps, drops and noise are shown in Supplementary Fig. 19a. Flaps produce pseudo-periodical deviations in total acceleration with negative and positive deviations of approximately similar magnitude. These almost sinusoidal deviations are produced by regular up–down wing movements. Contrary to flaps, drops are characterized by a strong negative deviation followed by a slow positive compensation. They are produced by momentary folding of one or both of the wings (see Supplementary Movie 4). Noise is characterized by smaller deviations around the zero time point and on average has a symmetrical shape (relative to the zero time point). The distribution densities of the maximal deviation of acceleration (at zero time) shown in Supplementary Fig. 19b demonstrate that flaps can be readily separated from noise by simply selecting a threshold around $0.6g_0$. However, separation of drops from flaps and noise required information about the signal shape. To estimate the duration of flapping flight we summed the $\pm 0.35$ s interval around flap detection points.

**Wind speed analysis.** Wind information (absolute value and direction at the birds' location) was obtained from the Movebank database (www.movebank.org). The database provides wind speeds for altitudes $>100$ m. For lower altitudes between 10 and 100 m, wind speed was computed from the wind data at 10 m using the equation $W = W_{10}(h/h_{10})^\alpha$, where $W$ is the wind speed at the desired altitude $h$; $W_{10}$ the known wind speed at altitude $h_{10} = 10$ m over mean sea level; and $\alpha$ the Hellmann exponent. In this study, the Hellmann exponent ($\alpha = 0.03958$) was estimated from the average ratio of wind speed at altitudes 100–150 m (given by the database) to $W_{10}$. Because the altitudes given by GPS are not precise enough to be used for calculating wind speed at altitudes $<10$ m, the value $W_{10}$ was taken as an estimate of wind speed.

**Statistical analysis.** For comparisons between flight and land $N = 9$, whereas for in-flight comparisons $N = 14$. Unless stated otherwise, reported values are the mean $\pm$ s.e.m. Paired Student's $t$-tests (two-tailed) were used in most cases. Quantities expressed as a per cent were first normalized using a Fisher transformation. For the time course of SWA and the different types of SWS on land, the analysis was restricted to birds with at least 10 h of recording time ($N = 7$), and only the first 10–12 h were used for this analysis. For the analysis of the relationship between sway acceleration and EEG asymmetry (Fig. 3), the mean of the sway values $>0.175g_0$ (to left) and $<-0.175g_0$ (to right) for individual birds were used.

**Data availability.** The authors declare that the data supporting the findings of this study are available within the article and its Supplementary Information Files, or from the corresponding authors upon request.

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

## Acknowledgements

We thank the Galápagos National Park Service for permission to work in the park, the Charles Darwin Research Station for logistical support and the Charles Darwin Foundation. We thank Peter Abegg for veterinary supervision, Martina Oltrogge for help with scoring the accelerometry, Andrei Abramchuk and Irina Panova for help with the Neurologger 2A development, Ninon Ballerstädt for drawing the skull, Matthias C. Berger for making Supplementary Movies 2 and 3 and Dolores Martinez-Gonzalez for comments on the manuscript. The Max Planck Society, the University of Zurich and the Swiss National Science Foundation supported the project.

## Author contributions

N.C.R. and A.L.V. designed the study. H.-P.L. and M.W. were involved in the early development of the project. S.M.C., G.D'O., N.C.R., R.T. and B.V. collected data. N.C.R. and A.L.V. analysed the EEG data. N.C.R., B.V. and A.L.V. analysed the accelerometry and GPS data. A.L.V. performed the statistical analyses. S.C., N.C.R. and A.L.V. created the figures. N.C.R. and A.L.V. wrote the paper with input from all coauthors.

## Additional information

**Competing financial interests:** The authors declare no competing financial interests.

