## [Peer review file · Nature Communications]

Reviewers' Comments:

Reviewer #1 (Remarks to the Author)

A. Summary of the key results

I would like to congratulate the authors on completing this excellent and difficult study, as well as their analysis and very well written MS. The study showed that frigatebirds slept in flight for only 5% of 24-h, 1/10th of the time spent sleeping on land, virtually sacrificing sleep in exchange for vigilance. Another important finding reveals that birds displayed both bilaterally symmetrical and asymmetrical SWS during flight. The data on sleep in frigatebirds support the idea that asymmetrical SWS universally serves sleeping and monitoring the environment simultaneously as some other birds and mammals do and the notion of the great importance of ecological demands in the regulation of sleep in mammals and birds, which significantly impacts the understanding of the function of sleep.

B. Originality and interest: if not novel, please give references

This is the first recording of sleep in birds in flight. I don't think there were doubts that birds sleep while engaging in multi day flight, but this did need to be documented and quantified someday.

C. Data & methodology: validity of approach, quality of data, quality of presentation

As I mentioned in Summary the authors have done excellent job and extensive analysis. With that being said, I have some reservations regarding the analysis completed at this stage and would like to request additional information. Please see F.

D. Appropriate use of statistics and treatment of uncertainties

In my view, all stat treatments were done appropriately.

E. Conclusions: robustness, validity, reliability

I have one major concern specified in chapter F. Except for this all other conclusions are very well explained and supported.

F. Suggested improvements: experiments, data for possible revision

I have one major concern:

Line 259 states that only one 24-h period was scored in each frigate bird during flight. The next sentence further details that "the day was selected to maximize the time since instrumentation, signal quality, and the amount of sleep." All of this need some additional information, clarification and some discussion.

I don't see specific criteria for this day selected except for the "signal quality." My concern is to how one selected day represents the daily variation and average amount of sleep over the whole foraging trip?.

In addition, different birds spent on the wing a different number of days. If the scoring day was selected to "maximize the time since instrumentation," does this imply that this was the last day before landing? If my understanding is correct, these data show how the birds slept on the last day of their trip only.

It would be even more difficult to understand if "the amount of sleep" was used as a criterion to select the day to be scored.

It was shown (including the authors' prior publications) that animals may not sleep as regularly during periods of migration as opposed to the nonmigratory state. They can even go without sleep for a few days.

I fully acknowledge the difficulty recording EEG in flying birds. However, in my opinion, several consecutive days would have to be scored in one bird minimum (or in a few birds if possible) to gain reliable information on how the amount of sleep varied over these days in flight. This would be of a huge scientific value. Another approach is to score several days in each bird and to present the data for certain time periods relative to the beginning of the flight period.

To conclude, in my view considering the birds spent up to 10 days in flight, analyzing one out of 10 days is insufficient to present the whole story, which entails characterizing the amount of time frigate birds sleep during flight. As I mentioned above I think all of this need some additional information, clarification and some discussion.

Minor concerns:

Line 73. The descriptive information on lines 72-74 could be supplemented by the duration of continuous recording and how the birds were split in terms of the duration of recording in flight.

Line 79. Some estimate of the time birds engaged feeding / or some other behaviors would be interesting to add in addition to the data on how much time they spent flapping, ascending and circling without referencing the activity.

Lines 112-113. "Episodes of SWS were longer during circling flight (14.53 ± 1.24 s) than straight flight (6.74 ± 0.38 s, $P=3.0 \times 10^{-5}$)." The reported SWS episode durations are short. It does not agree with the apparent duration of SWS episodes shown in all figures which lasted minutes creating confusion. How was the duration of episodes calculated and what meaning does it bear?

Lines 160 -162. I am confused on the large difference between average (28 sec on land) and max (120-500 sec) SWS durations (5-20 folds), likely meaning that the majority of SWS episodes were very short episodes lasting seconds. I am getting a different point of view when looking at the figures showing long periods of SWS lasting minutes.

Lines 173-183. The last paragraph of the Results section suggests that sleep is homeostatically regulated in frigatebirds. This is a particularly important issue considering an apparent dramatic reduction of sleep time in the frigates during the in-flight period. Considering "the landing time and recording duration varied across birds, the sample size for the respective hours varies between 3 and 9" (supplement data, Fig. 5), I doubt the data presented in the last results paragraph (including Fig 4) allow any interpretation at this point. My opinion is that the data on recovery sleep after landing is sketchy and largely missing. However, these are objective difficulties for which the authors cannot be blamed.

G. References: appropriate credit to previous work?
All important prior studied are referenced and credited.

H. Clarity and context: lucidity of abstract/summary, appropriateness of abstract, introduction and conclusions
This MS is very well written.

Reviewer #2 (Remarks to the Author)

This is a very fascinating study breaking new ground by demonstrating for the first time that birds can sleep in flight. I wish to congratulate the authors warmly on this impressive achievement! The possible occurrence of sleep was investigated for female great frigatebirds (n=14) making foraging flights over the Pacific Ocean lasting on average 5.8 d (flights during day and night without landing). Neurophysiological activity was recorded by EEG from the hyperpallium, and simultaneously accelerometer loggers recorded flight modes (wing beat patterns, gliding and circling flight) and head movements, and GPS data loggers recorded overall movements and altitudes. These data have been combined into a fascinating analysis and exploration of the flight behaviour of the frigatebirds and their sleep during flight compared to sleep while at the nest on land.

An impressive series of novel and exciting results are reported, e.g.: (1) Sleep occurred much more frequently when the birds were on land (53% of total time) than in flight (only 6% of time). (2) Sleep in flight took place mainly during the night (11% of night-time but only 0.5% of daytime) while sleep was regular during both night (54% of night-time) and day (48%) when on land. (3) Sleep was more often asymmetric (difference between the cerebral hemispheres) in flight (72% of sleep time) than on land (48%). (4) Sleep was also less intense (Fig 4b) and taking place in shorter episodes in flight (average duration of sleep episodes was 14 s) than on land (52 s). (5) Sleep in flight was often associated with circling/soaring flight, with the hemisphere in the direction of circling sleeping, while the other hemisphere was awake and receiving visual information from the eye in the direction of circling (fig. 3). (6) Sleep intensity tended to be highest soon after landing showing a gradual decline during the hours after landing. This is interpreted as postflight recovery sleep, but it remains to be investigated if it compensates fully or not for the limited sleep obtained in flight. (7) The frigatebirds were observed to sometimes climb to high altitudes, exceeding 600 m asl, during short periods mainly in the late afternoon. Sleep did occur during these ascents, but not always and sleep also occurred at lower altitudes, indicating that these ascents were not exclusively performed to facilitate sleep (other possible purposes may have associated with feather preening in flight and gaining environmental information).

I was much impressed by this multifaceted exploration of the occurrence of sleep among the frigatebirds. I read the manuscript with greatest interest and pleasure, with its concise and clear information/explanations and elegant and informative illustrations (also in Supplement).

I have no important suggestions of revision. However it may be useful to have the main results briefly listed in a table or text box to facilitate for the reader to find all fascinating novel aspects that are addressed and revealed in the study (e.g. the seven novel findings exemplified above and possibly others). Furthermore it would be interesting to know the fate of the experimental birds - were all of them released at full health after the anaesthesia and removal of instruments?

Please note that my expertise is in behavioural ecology (migration and flight behaviour/performance in birds) and I have no neurophysiological competence to critically inspect and validate the classifications and durations of sleep (SWS and REM) from the EEG recordings. As pointed out above, this study is opening new ground in behavioural ecology by demonstrating sleep in flight and initiating novel analyses of the behavioural and ecological implications associated with sleep in free-living animals. The authors have an eminent experience in these methods and in neurophysiology, are world-leading in the field of animal sleep, and the explanations and results in the manuscript are carefully described and very convincing. Still, the EEG patterns and results should be examined by a reviewer with special expertise in neurophysiology.

Again, I wish to congratulate the authors warmly on this impressive and multifaceted study demonstrating that birds can sleep on the wing!

Lund 8 March 2016

Thomas Alerstam

NCOMMS-16-04103: Response to Referees

Our responses are in bold italicized text.

Reviewer #1 (Remarks to the Author):

A. Summary of the key results

I would like to congratulate the authors on completing this excellent and difficult study, as well as their analysis and very well written MS. The study showed that frigatebirds slept in flight for only 5% of 24-h, 1/10th of the time spent sleeping on land, virtually sacrificing sleep in exchange for vigilance. Another important finding reveals that birds displayed both bilaterally symmetrical and asymmetrical SWS during flight. The data on sleep in frigatebirds support the idea that asymmetrical SWS universally serves sleeping and monitoring the environment simultaneously as some other birds and mammals do and the notion of the great importance of ecological demands in the regulation of sleep in mammals and birds, which significantly impacts the understanding of the function of sleep.

We thank the reviewer for their supportive comments.

B. Originality and interest: if not novel, please give references

This is the first recording of sleep in birds in flight. I don't think there were doubts that birds sleep while engaging in multi day flight, but this did need to be documented and quantified someday.

We agree that most people assumed that birds can sleep in flight. In particular, it was commonly assumed that they had to sleep unihemispherically in flight. As such, one unanticipated result from our study is that in addition to sleeping unihemispherically, frigatebirds can also sleep bihemispherically in flight. However, perhaps the most remarkable finding from our study is that despite being able to engage in all types of sleep in flight, frigatebirds actually sleep remarkably little on the wing. This unanticipated finding challenges the dominant view that a substantial amount of sleep is needed every day to maintain adaptive cognitive performance. Consequently, our study underscores the importance of testing commonly held assumptions regarding sleep in animals faced with challenging ecological demands in the wild.

C. Data & methodology: validity of approach, quality of data, quality of presentation

As I mentioned in Summary the authors have done excellent job and extensive analysis. With that being said, I have some reservations regarding the analysis completed at this stage and would like to request additional information. Please see F.

We have addressed this important suggestion from the reviewer below.

D. Appropriate use of statistics and treatment of uncertainties

In my view, all stat treatments were done appropriately.

Great!

E. Conclusions: robustness, validity, reliability

I have one major concern specified in chapter F. Except for this all other conclusions are very well explained and supported.

F. Suggested improvements: experiments, data for possible revision

I have one major concern:

Line 259 states that only one 24-h period was scored in each frigate bird during flight. The next sentence further details that "the day was selected to maximize the time since instrumentation, signal quality, and the amount of sleep." All of this need some additional information, clarification and some discussion.

I don't see specific criteria for this day selected except for the "signal quality." My concern is to how one selected day represents the daily variation and average amount of sleep over the whole foraging trip?.

In addition, different birds spent on the wing a different number of days. If the scoring day was selected to "maximize the time since instrumentation," does this imply that this was the last day before landing? If my understanding is correct, these data show how the birds slept on the last day of their trip only.

It would be even more difficult to understand if "the amount of sleep" was used as a criterion to select the day to be scored.

It was shown (including the authors' prior publications) that animals may not sleep as regularly during periods of migration as opposed to the nonmigratory state. They can even go without sleep for a few days.

I fully acknowledge the difficulty recording EEG in flying birds. However, in my opinion, several consecutive days would have to be scored in one bird minimum (or in a few birds if possible) to gain reliable information on how the amount of sleep varied over these days in flight. This would be of a huge scientific value. Another approach is to score several days in each bird and to present the data for certain time periods relative to the beginning of the flight period.

To conclude, in my view considering the birds spent up to 10 days in flight, analyzing one out of 10 days is insufficient to present the whole story, which entails characterizing the amount of time frigate birds

sleep during flight. As I mentioned above I think all of this need some additional information, clarification and some discussion.

We agree that our description of how the day in flight was selected for analysis was unclear. In addition, we agree that the paper would be strengthened greatly by quantifying the time spent sleeping across more than one day of flight. Consequently, we have now quantified sleep for as many days as possible. As a result, our sample size has increased from 14 days to 70 days in flight. We now present the mean amount of time spent sleeping across the flight, and show the day-day variation in sleep for all 14 birds (Supplementary Figs. 1-8). In addition, in these plots we have also marked the days with large amounts of sleep that were selected for scoring in the initial submission. Because we focused on days with maximal sleep in the initial submission, the mean daily sleep duration has decreased from 1.4 to 0.7 hours per day. As such, this further emphasizes the most unanticipated aspect of our results—despite being able to engage in all types of sleep in flight, frigatebirds sleep remarkably little for extended periods of time.

Minor concerns:

Line 73. The descriptive information on lines 72-74 could be supplemented by the duration of continuous recording and how the birds were split in terms of the duration of recording in flight.

The duration of all recordings and flights is now shown along with the sleep scoring in Supplementary Figs. 1-8.

Line 79. Some estimate of the time birds engaged feeding / or some other behaviors would be interesting to add in addition to the data on how much time they spent flapping, ascending and circling without referencing the activity.

We have now elaborated on our description of flight behavior in Supplementary Fig. 9a-c. The analysis now includes the number of flaps and drops, the time spent flapping and circling, as well as the time spent below 20 m, a proxy for potential feeding. In addition, in Supplementary Fig. 11, we show that there was no bias for circling to the left or right.

Lines 112-113. "Episodes of SWS were longer during circling flight (14.53{plus minus}1.24 s) than straight flight (6.74{plus minus}0.38 s, $P=3.0\times 10^{-5}$)." The reported SWS episode durations are short. It does not agree with the apparent duration of SWS episodes shown in all figures which lasted minutes creating confusion. How was the duration of episodes calculated and what meaning does it bear?

The reviewer is correct that it was unclear how we calculated sleep bout durations. We have now added a sentence clarifying this method:

Line 280: "A bout of a given state was defined as one or more epochs of that state uninterrupted by a single epoch of another state."

As a result, a single 4-s epoch of another state (wake or REM sleep) would terminate a bout of SWS.

Also, because the primary aim of this study was to determine whether birds can sleep in flight, we focused on rather pronounced examples in Fig. 2 and 3. We now state this in the caption for Fig. 2:

“The mean duration of episodes of sleep was shorter than these long episodes used to demonstrate USWS and BSWs in flight.”

Lines 160 -162. I am confused on the large difference between average (28 sec on land) and max (120-500 sec) SWS durations (5-20 folds), likely meaning that the majority of SWS episodes were very short episodes lasting seconds. I am getting a different point of view when looking at the figures showing long periods of SWS lasting minutes.

Yes, the reviewer is correct that episodes of sleep are usually rather short, as shown in the mean data. For the reason stated above, we focused on long episodes of sleep in the figures. We now state this in the caption for Fig. 2:

“The mean duration of episodes of sleep was shorter than these long episodes used to demonstrate USWS and BSWs in flight.”

The mean data shows how the birds typically sleep, and the maximum duration data in the text shows what they are capable of doing.

Lines 173-183. The last paragraph of the Results section suggests that sleep is homeostatically regulated in frigatebirds. This is a particularly important issue considering an apparent dramatic reduction of sleep time in the frigates during the inflight period. Considering "the landing time and recording duration varied across birds, the sample size for the respective hours varies between 3 and 9" (supplement data, Fig. 5), I doubt the data presented in the last results paragraph (including Fig 4) allow any interpretation at this point. My opinion is that the data on recovery sleep after landing is sketchy and largely missing. However, these are objective difficulties for which the authors cannot be blamed.

As the reviewer dully notes, we would have liked to have had more recovery sleep data. However, we feel that there is more than sufficient data to establish that homeostatic processes were operating on land. We think that the way we presented the data may have been unclear. The data can be presented in either of two ways; 1) aligned with the time the bird landed as time zero on the x-axis, or 2) aligned relative to the actual clock time. We chose the second option in the initial Supplementary Fig. 5. As a result, the first few hours have relatively few birds contributing to the means for those hours. However, as stated in the paper, in the 9 birds with post-flight recordings, we obtained on average 16.4 hours, with 7 of these lasting more than 10 hours. In previous studies of sleep homeostasis in mammals and birds 10 hours is more than enough time to establish that EEG slow wave activity (a marker for this homeostatic process) progressively declines during recovery sleep following a period of sleep loss. Indeed, our statistical tests show that this is also clearly the case in our birds. Consequently, we are confident that we have established that there is a homeostatic response to sleep loss in flight. Nonetheless, as acknowledged in the initial paper, we are unable to fully characterize the extent of this recovery. As recognized by the reviewer, this would require recordings lasting up to 20 days which is beyond the capacity of batteries small enough to be carried on the bird's head.

Given this discussion, we now feel that it is important to present the recovery data in both formats; aligned to the clock as before, and aligned to the time from landing. Both are now shown in Supplementary Fig. 9e,f.

G. References: appropriate credit to previous work?
All important prior studies are referenced and credited.

Great!

H. Clarity and context: lucidity of abstract/summary, appropriateness of abstract, introduction and conclusions

This MS is very well written.

Thank you.

Reviewer #2 (Remarks to the Author):

This is a very fascinating study breaking new ground by demonstrating for the first time that birds can sleep in flight. I wish to congratulate the authors warmly on this impressive achievement!

The possible occurrence of sleep was investigated for female great frigatebirds (n=14) making foraging flights over the Pacific Ocean lasting on average 5.8 d (flights during day and night without landing).

Neurophysiological activity was recorded by EEG from the hyperpallium, and simultaneously accelerometer loggers recorded flight modes (wing beat patterns, gliding and circling flight) and head movements, and GPS data loggers recorded overall movements and altitudes. These data have been combined into a fascinating analysis and exploration of the flight behaviour of the frigatebirds and their sleep during flight compared to sleep while at the nest on land.

An impressive series of novel and exciting results are reported, e.g.: (1) Sleep occurred much more frequently when the birds were on land (53% of total time) than in flight (only 6% of time). (2) Sleep in flight took place mainly during the night (11% of night-time but only 0.5% of daytime) while sleep was regular during both night (54% of night-time) and day (48%) when on land. (3) Sleep was more often asymmetric (difference between the cerebral hemispheres) in flight (72% of sleep time) than on land (48%). (4) Sleep was also less intense (Fig 4b) and taking place in shorter episodes in flight (average duration of sleep episodes was 14 s) than on land (52 s). (5) Sleep in flight was often associated with circling/soaring flight, with the hemisphere in the direction of circling sleeping, while the other hemisphere was awake and receiving visual information from the eye in the direction of circling (fig. 3).

(6) Sleep intensity tended to be highest soon after landing showing a gradual decline during the hours after landing. This is interpreted as postflight recovery sleep, but it remains to be investigated if it compensates fully or not for the limited sleep obtained in flight. (7) The frigatebirds were observed to sometimes climb to high altitudes, exceeding 600 m asl, during short periods mainly in the late afternoon. Sleep did occur during these ascents, but not always and sleep also occurred at lower altitudes, indicating that these ascents were not exclusively performed to facilitate

sleep (other possible purposes may have associated with feather preening in flight and gaining environmental information).

I was much impressed by this multifaceted exploration of the occurrence of sleep among the frigatebirds. I read the manuscript with greatest interest and pleasure, with its concise and clear information/explanations and elegant and informative illustrations (also in Supplement).

I have no important suggestions of revision.

We thank the reviewer for his enthusiastic review of our manuscript!

However it may be useful to have the main results briefly listed in a table or text box to facilitate for the reader to find all fascinating novel aspects that are addressed and revealed in the study (e.g. the seven novel findings exemplified above and possibly others).

We have added section headings to help the readers anticipate what follows. In addition, as stipulated by Nature Comm, we have also added a summary paragraph at the end of the Introduction. We will leave further changes to the discretion of the editors.

Furthermore it would be interesting to know the fate of the experimental birds - were all of them released at full health after the anaesthesia and removal of instruments?

As is evident from our high return rate (14/15 birds) our procedures did not appear to have a large impact on the birds. Following removal of the equipment the birds appeared in good health and their nesting behavior was indistinguishable from that of the other birds.

We have now added a sentence stating this.

Line 270: "Upon release, the birds resumed nesting behavior indistinguishable from that observed in undisturbed birds."

Also, we recently learned that a colleague not involved in this project later worked in this colony, and observed our birds exhibiting normal parental behavior.

Please note that my expertise is in behavioural ecology (migration and flight behaviour/performance in birds) and I have no neurophysiological competence to critically inspect and validate the classifications and durations of sleep (SWS and REM) from the EEG recordings. As pointed out above, this study is opening new ground in behavioural ecology by demonstrating sleep in flight and initiating novel analyses of the behavioural and ecological implications associated with sleep in free-living animals. The authors have an eminent experience in these methods and in neurophysiology, are world-leading in the field of animal sleep, and the explanations and results in the manuscript are carefully described and very convincing. Still, the EEG patterns and results should be examined by a reviewer with special expertise in neurophysiology.

Again, I wish to congratulate the authors warmly on this impressive and multifaceted study demonstrating that birds can sleep on the wing!

Lund 8 March 2016

Thomas Alerstam

Thanks, again!

Reviewers' Comments:

Reviewer #1 (Remarks to the Author)

The authors have edited the MS answering to my concerns. I don't have other questions. My recommendation it to accept the MS for publication.